# Preliminary Investigation of the Usability Characteristics Required for Wound Management Products by Semi-Structural Interview of Medical Staff

**DOI:** 10.3390/pharmacy8030152

**Published:** 2020-08-22

**Authors:** Kaoru Hirose, Yayoi Kawano, Nahoko Shigeno, Yoshikatsu Mizutani, Hiraku Onishi, Takehisa Hanawa

**Affiliations:** 1Faculty of Pharmaceutical Sciences, Tokyo University of Science, 2641, Yamazaki, Noda-city, Chiba 278-8510, Japan; sola_aoi@hotmail.com (K.H.); y.kawano@rs.tus.ac.jp (Y.K.); pri75beauty@gmail.com (N.S.); 3a14101@ed.tus.ac.jp (Y.M.); 2Department of Drug Delivery Research, Hoshi University School of Pharmacy and Pharmaceutical Sciences, 2-4-41, Ebara, Shinagawa-ku, Tokyo 142-8501, Japan; onishi@hoshi.ac.jp

**Keywords:** wound management product, usability, pressure ulcer, thematic analysis, semi-structured interview

## Abstract

Consideration of drug usability characteristics is important during the design process. Although many wound management products have been developed in recent years, there are few studies on their usability. We investigated the needs and characteristics of wound management products required by medical professionals, so as to consider these in future development projects. Semi-structured interviews were conducted in a group of healthcare professionals. Interview responses were analyzed based on thematic analysis. Four themes common to all facilities were secondary wounds, adaptability of materials, convenience, and physicochemical properties. Economic efficiency of medical care was found to be considered only at the hospital, and quality of life of patients was found to be considered only at the home palliative care clinic. Requirements for wound management products can be affected by participants’ roles and their facility settings. However, there were needs common to all fields that all wound management products should aim to incorporate.

## 1. Introduction

Chronic wounds such as pressure ulcers (PU) and diabetic gangrene are often difficult to treat. They can also trigger infectious diseases and affect underlying diseases and cause considerable human and economic human costs in treatment [1,2]. Leg ulcers associated with deep vein thrombosis, diabetes, and arteriosclerosis obliterans can lead to amputation of a leg, as well as impairing the activities of daily living or quality of life (QOL) of patients and increasing the burden on caregivers [3]. Generally, chronic wounds are treated with topical products (ointments, creams, or sprays); wound dressings; and negative-pressure wound therapy (NPWT) [4,5,6,7]. Since the concept of moist wound healing (MWH) was established, there has been a remarkable increase in the availability of wound dressings that maintain a moist environment [8]. Wound dressings with effects such as antibacterial silver and pressure reduction based on their thickness have also been developed [2,9,10,11]. Dressings are easy to use and can be applied by family and caregivers, as well as healthcare professionals. However, they are more expensive than topical products [12]. Meanwhile, no new topical products have come into wide use in Japan since Iodocoat^®^ was released in 2005. Iodocoat^®^ simplifies the cleaning of wound surfaces, because it does not remain in the wound surface by absorbing exudates into a gel matrix [13]; then, the characteristics of the formulation are very useful when changing them at the bedside. 

In recent years, the blending of two topical products and treatment methods for the purpose of promoting wound healing, known as the Furuta method, has become well-known among pharmacists engaged in PU treatment in Japan. The Furuta method aims to maintain the moist conditions of wound surfaces by combining topical formulations, such that the active ingredients can be reliably delivered to wound surfaces via appropriate application. Topical formulations are composed of 90% or more of base materials, classified based on intended purposes such as moisturizing and water absorption. In the Furuta method, the mixing of two topical products containing the right ointment base enables the water absorption and/or retention to maintain the desired level of moisture at the wound. This approach is cheaper than dressings, and water absorption and refilling can be adjusted by preparing blends of topical products accordingly [14].

However, healthcare professionals practicing the Furuta method require a great deal of expertise and experience in PU treatment due to the complexity of selecting and blending the correct products. The usability of products for patients or caregivers is an important factor in drug development. Wound management products (WMP) should be as user-friendly as possible, so that all care providers, including patients themselves, can understand and apply them. To the best of our knowledge, although there have been studies on the usability of existing topical products and dressings [15,16,17], the functionality and usability required for WMPs have not been evaluated. This information would help in future designs of new WMPs.

We used a qualitative research approach to investigate the usability, so as to utilize non-numerical data. This approach was developed to elucidate phenomena that are not well-suited for experiments and statistics and is applied to medical science and nursing research [18,19]. Here, a small number of opinions can be evaluated in a manner ruled out by a quantitative approach [20]. We used this approach to interview healthcare professionals engaging in PU treatment and evaluated their needs regarding the usability of WMPs.

## 2. Materials and Methods 

Interviews were conducted with 14 healthcare professionals managing PU between 1 April 2018 and 30 June 2019. Facilities and participants were selected based on purpose-oriented sampling, where participants were medical doctors, nurses, pharmacists, or dietitians who worked at a home palliative care clinic, university hospital, cardiovascular hospital, or general hospital and were engaged in the treatment of chronic wounds at least once a month. The characteristics of the participants and interview time are depicted in Table 1. A pharmacist who works at a school of pharmacy and a pharmacy student conducted semi-structured interviews with 1 to 3 people at a time at each facility, based on an interview guide prepared in advance (Table 2). Comments during the interview were recorded and transcribed into verbatim records.

Data were analyzed based on an inductive thematic analysis (TA) [21]. At the first step, the recorded sounds were converted to text data; then, the obtained data were classified by categories. At the next step, the data obtained from the verbatim records were segmented (segmenting), and the contents of the segments were generated (coding). Datasets were compressed by coding, and codes were applied to the whole dataset to review its validity. Finally, the research theme was generated as the highest level of concept in the codes. We investigated in the clinic and hospitals which have different compositions of patients; then, it was regarded to reach the theoretical saturation in this study. Here, the coding and TA were performed using NVivo 12 Qualitative Data Analysis Software (QSR International, Victoria, Australia).

This research was approved by the Ethics Committee of the Tokyo University of Sciences, Tokyo, Japan. (approval number 17009, “Search for needs related to the user-friendliness of wound management products”.)

## 3. Results

### 3.1. Obtained Themes

As a result of coding, six themes and codes emerged: secondary wounds, adaptability of products, convenience, economy of healthcare, physicochemical properties, and patient QOL (Figure 1 and Table 3).

### 3.2. Secondary Wounds

The secondary wound theme included coded characteristics such as the ease of removal when changing WMPs, ease of cleaning wound surfaces, the need to check wound surfaces after applying dressings, and “range of application”. 

The ease of removal and cleaning refers to the reduction of pain when changing WMPs and skin tears in the fragile, exposed skin of elderly people. Interviewees reported that silicon products had a good balance between removal and adhesion and regarded them highly. Since checking wound surfaces is carried out daily to observe acute PU and infection, low-transparency dressings were reportedly avoided, especially for acute PU [22], so as to avoid having to removing dressings when making this inspection. In the range of applications, workers were concerned that side effects might occur due to exudates adhering to normal skin.

### 3.3. Adaptability

Adaptability included the considerations such as necrotic tissue removal, exudate control, infection control, narrow pockets or deep wounds, prevention, and “application to wide recovery stages”.

In necrotic tissue removal, participants preferred products that could remove necrotic tissue, especially by sloughing, without debridement. Although the absorption of exudates is required, the delaying of wound healing due to the excessive absorption of exudate and the sticking of products to wound surfaces are described as problems in exudate control. With regard to infection control, the study subjects suggested that treatment options seemed to be limited due to the types of products used for infected wounds. Since it is difficult to apply topical products to deep wounds with narrow pockets, participants suggested thin spray nozzles enabling medicines to be delivered, as well as products that can be filled into a deep wound. Further, since the delivery and retention of topical products to wound surfaces affects the healing, some participants commented that formulations to promote wound healing are needed. In prevention, participants wanted WMPs that prevent against medical device-related pressure ulcers (MDRPUs) and skin tearing. Both MDRPUs and skin tears are relatively new wound concepts that can be prevented. The problem caused by the use of therapeutic WMPs as prevention is also related to the “economy of medical care” (described below). In terms of adaptation for wide recovery stages, “REF-THERA,” which is a mixture of Reflap^®^ ointment and Theradia^®^ paste, was mentioned. REF-THER absorbs exudates and has an antibacterial effect and is often used for shallow PUs occurring in home care. In addition, there was an opinion from participants working in the hospital that selecting products with a wide range of adaptability will reduce the number of required products.

### 3.4. Convenience

The theme of convenience was mentioned, with ease of use and formulation being important when using the WMP. For instance, sprayed products were described as difficult to use, but solid products were described as easy to use. Participants tended to feel that it would be convenient if solid products were individually packaged. However, there was also an opinion that solid products such as Geben^®^ cream and pastes are too solid and, therefore, cannot be used straight from the tube. Sheets and patches made a good impression, where “hydrogel sheets including active pharmaceutical ingredients that can be applied simply are easy and can be used by the patient/family”, but participants also felt that “the adhesive surfaces stick together when they are applied”. In an interview at the general hospital, different medical personnel may treat wounds for the same patient, and there was concern that the method of applying dressings and the amount of topical products can vary. As a result, participants desired products that could be used uniformly by all users. Furthermore, because it can be difficult for nursing staff to wash away products, participants preferred easily washed macrogol bases such as Isodine^®^ gel. Therefore, convenience at the time of washing appears to be important. In interviews at the home palliative care clinic, we found that elderly people without access to the internet or telephone had difficulty in ordering dressings and that convenience in purchasing WMPs is also required.

In the theme of usage at both the hospital and home palliative care clinic, participants noted that extended treatment times and an increasing number of treatments impose a burden on patients. They felt that shortening the treatment time and reducing the number of treatments would reduce these burdens. In terms of preservation and management, there was a desire to store WMPs at the patient’s bedside for easy access, with cold storage requirements believed to be troublesome and wasteful. Participants cited experiences of WMPs deteriorating as a result of being left at room temperature or re-prescribing them because they could not be immediately found. As a result, participants indicated that cold-stored WMPs were often avoided in clinical settings. Similarly, at the home palliative care clinic, participants felt that many patients leave products under hot and humid conditions and suggested a need for products that were not affected by storage conditions.

### 3.5. Economy of Medical Care

Participants expressed opinions on the economy of medical care only in the hospital setting. Here, participants suggested a preference for dressings that could be temporarily removed to confirm wound surface conditions, then reattached, so as to avoid repeatedly having to apply new and costly dressings. Additionally, they noted that many families choose ointments and gauze for treatment rather than buying dressings twice a week due to cost, suggesting that the use of dressings may put pressure on caregivers’ budgets. For standard products adopted for use in hospitals, there was a tendency to choose WMPs with a wide range of applications to limit the number of products required. As a result, participants placed importance on inexpensive WMPs. In addition, nurses who mentioned insurance coverage indicated a desire for the coverage of WMPs intended for MDRPU prevention and skin tearing. Finally, participants described often discarding products with short expiration periods, such as Fiblast^®^ spray, which they felt was wasteful. 

### 3.6. Physicochemical Properties

Physicochemical properties reflected the properties and shapes required for WMPs. Thickness and hardness were most frequently discussed by participants, with the feeling that dressings should be as thin as possible so as to avoid interfering with daily life. However, participants did indicate the need for some thicker options, such as for sacral dressings that might be compressed when sitting. Many comments discussed the hardness of topical product bases. Hardness differs depending on the base materials and directly affects the ease of use. We suggest that the hardness of base materials can affect the treatment, so topical products are required to have an appropriate hardness.

Dressings that can be cut and applied to make contact with wound surfaces, as required in any large PU, are necessary in dealing with a wide variety of wounds. It is important to maintain the form and adhesion after application of the product to the wound surface. If a product is displaced when moving or sitting, drug delivery to the wound is inhibited, and healing is delayed. Therefore, high adhesion is required in WMPs, but if the adhesion is too great, as mentioned in the theme of secondary wounds, products may cause epidermal peeling when they are exchanged. Therefore, WMPs with a good balance between adhesion and removal are appreciated.

Finally, participants mentioned that, when topical products dripped too readily when treating supine patients, they were apt to cause skin inflammation and secondary wounds. 

### 3.7. QOL of Patients

The QOL of patients was discussed only in the home palliative care clinic. Participants felt that the number of treatments should be smaller in the field of home healthcare, where manpower is more limited than in a hospital. There was also a comment that patients felt embarrassed to have their clothes soiled due to topical product leakage, suggesting that such failures may affect the dignity of patients and their willingness to be treated. 

## 4. Discussion

The common themes obtained at all facilities included secondary wounds, product adaptability, convenience, and physicochemical properties. Since wound treatment takes time and manpower, highly convenient WMPs are likely to be required in a number of medical settings. In particular, highly stable and individually packaged sheet-like products tended to be preferable, since they can be used uniformly by all workers. WMPs that are easy to clean to reduce the treatment time are also desirable, since both trained medical personnel and general caregivers lacking specialist knowledge may be tasked with handling WMPs. Similarly, since wounds are treated daily or every few days, participants were concerned that the stability and hygiene of the WMPs may be affected depending on the storage conditions of the WMP in the patient’s home. As a result, we identified a need for products that can be as easily applied by caregivers as medical professionals, in order to achieve the same treatment outcome every time. 

The effects of the required products differed depending on the wound stage, especially for PU, and followed the four key themes of necrotic tissue removal, exudate control, infection, and deep wounds with narrow pockets. In the Japan Society of Pressure Ulcers guidelines, different topical products/dressings are recommended according to each stage [21]. However, participants commented on the wide range of products on the market, with many that are similar and difficult to choose between. This suggests that even medical staff who have profound knowledge about the care for PU find it difficult to select the appropriate products.

Participants indicated that stage-independent WMPs are desired. “REF-THERA is moderately water-absorbing, and can granulate, epithelialize, and control infection, and is therefore preferable.” Currently, only Bromelain ointment is covered by insurance for the removal of necrotic tissue. The Japan Society of Pressure Ulcers guideline also recommends Geben^®^ cream, Cadex^®^ ointment, Debrisan^®^ paste, and similar products for debridement [22]. Since it is mainly composed of proteolytic enzymes, bromelain ointment tends to cause side effects on normal skin. Therefore, when using it, it is necessary to protect the wound periphery with white petrolatum before use, which complicates the procedure.

Although surgical debridement was mentioned as a part of wound care, participants commented that it is better that sloughed materials are dissolved and removed and that necrotic tissue should be removed without debridement. As a result, there was a preference for chemical debridement with topical products rather than surgical debridement. Although chemical debridement causes less bleeding and pain than surgical debridement, it takes a longer time to accomplish. In weighing risks and benefits when selecting MWPs, reducing the time required for debridement was a desired function for personnel selecting chemical debridement.

For the management of exudate, there are many dressings on the market aimed at absorbing and retaining exudate based on MWH, which participants commented on. However, the absorptive capacity of WMPs differs depending on their physical properties [23]. Therefore, if the balance between the volume of exudate and absorptive capacity of WMP is lost, the number of product replacements may be increased or the wound surface may become dry, delaying wound healing. Accordingly, MWP with adequate absorptive capacity and usefulness for a long time would be desirable.

Adhesion as a physicochemical property is also linked to the theme of secondary wounds. Skin tears, or traumatic wounds occurring by shearing and friction forces, have become points of concern for elderly patients with reduced skin elasticity and strength. Tears can also occur due to tape used to fix indwelling needles or gauze [24,25]. WMPs with strong adhesion prevent dressing slippage and can fit highly mobile sites but may cause skin damage and pain when products are exchanged. Furthermore, when gauze sticks to wound surfaces, bleeding may occur when dressings are changed. Therefore, it is necessary to strike an appropriate balance between adhesion and dressing removal.

Regarding the economy of the medical care, dressings were mentioned only in the hospital setting rather than the home palliative care clinic. Since dressings are more expensive than topical products, their use in applications such as MDRPU and the prevention of skin tears without insurance coverage may give the impression of financial loss for the hospital management. The Japanese healthcare system has universal insurance, and all patients bear part of the cost. The cost to patients varies depending on whether or not medical supplies and materials are covered by insurance. In other words, if dressings are used in a manner not covered by insurance, the hospital has to cover the whole cost. MDRPUs include all PUs caused by medical devices such as casts, elastic stockings, or endotracheal tubes [26]. In Japan, PUs caused by MDRPUs have been reported at various occurrence rates, such as 1.6% at a general hospital and 50% at a pediatric hospital [27]. Since skin tears often occur due to the adhesive tape used to fix the WMPs to PU wound surfaces, the prevention of this will help patients maintain their QOL and reduce medical costs, so demand is expected to increase in the future.

Further, in dressings intended to maintain moist environments, their application period tends to be longer. When dressings are removed due to natural separation or checking wound surfaces, a new one may be applied. Based on this, participants felt that reattachment characteristics were important. To overcome these drawbacks, dressings using silicone, which can be reapplied, have recently come into widespread use [28,29,30].

In the theme of patient QOL, obtained only in the home palliative care clinic, wound treatment was observed to be directly connected to daily life. Products that can reduce the number of required treatments can reduce the burden on patients and caregivers. Additionally, preventing the leakage of topical products contributes to maintaining patients’ QOL. Notably, although QOL likely should be considered at hospitals as well, we noted a difference between hospital settings that prioritize treatment and home care where treatment is performed in conjunction with daily life.

## 5. Conclusions

In this study, the required characteristics for WMPs are laid out and differ depending on the medical setting and background. However, there are common needs, and by verifying and incorporating these, it should be possible to develop WMPs that can be used in multiple fields. Moreover, based on the data obtained in this research, we should be able to construct an image of the developed product as it is used in the field and to show its orientation. 

This study is limited in having a small number of samples, but theoretical saturation was reached. Furthermore, since data from patients who are users and caregivers with little knowledge of wound care were not included, further information from patients and caregivers should be considered. In addition, the questionnaire survey based on the results of this study will be presented to medical personnel and caregivers for reflecting on the development of wound-healing products.

## Figures and Tables

**Figure 1 pharmacy-08-00152-f001:**
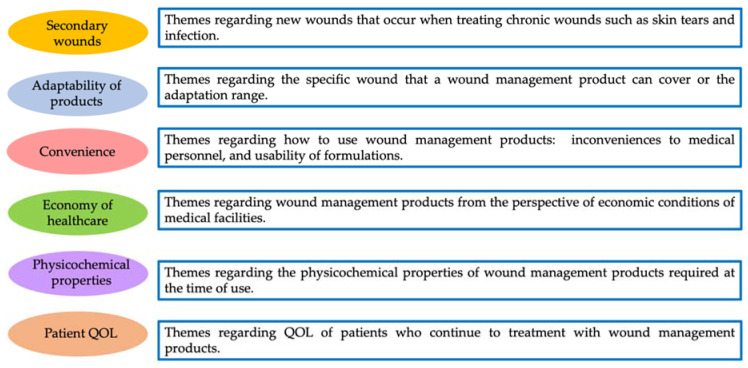
Themes obtained from the interviews. QOL: quality of life.

**Table 1 pharmacy-08-00152-t001:** Characteristics of interviewed participants.

	General Hospital 1 (H1)	General Hospital 2 (H2)	Cardiovascular Hospital (H3)	University Hospital (H4)	Home Palliative Care Clinic (HC)
Number of beds	200	292	88	610	–
Job category (Number)	WOC Nurse * (1)	Doctor (1) Pharmacist (1) WOC Nurse (1)	Doctor (1) Nurse (1) Pharmacist (1)	Doctor (1) WOC Nurse (1) Nurse (1) Dietician (1)	Doctor (1) Nurse (1) Pharmacist (1)
Duration of interview	1 h	1 h	1 h	1.5 h	1.5 h
Average of hospitalization period	13 days	14 days	6 days	11 days	–
Frequency of chronic wounds treatment (cases/month)	40–50	20–40	1–3	5–140	2–3
Types of chronic wounds	PU Ulcer due to deep vein thrombosis MDRPU Diabetic gangrene Arteriosclerosis obliterans Pyoderma gangrenosum Burn	PU Ulcer due to deep vein thrombosis MDRPU Burn	PU Refractory skin ulcer due to infection MDRPU	PU Ulcer due to deep vein thrombosis MDRPU Incontinence-associated dermatitis Venous stasis ulcer Skin inflammation due to stoma Burn	PU Skin trouble around the stoma Burn Cancerous tumor Diabetes-associated gangrene

* WOC Nurse: Wound, Ostomy and Continence Nurse.

**Table 2 pharmacy-08-00152-t002:** Interview guide.

Number	Question
1	How often are patients with chronic wounds treated each month?
2	What kind of chronic wounds have you been treating in the past?
3	What is the status of cooperation between occupation in the treatment of chronic wounds? And what are your approaches to treating chronic wounds in your facility?
4	What are the selection criteria for the WMPs you are using? Do you have any complaints or future expectations for WMPs?
5	What are the possible problems with applying the following products to chronic wounds?
6	What kind of consultation do you have with patients with chronic wounds and their families?

**Table 3 pharmacy-08-00152-t003:** The list of codes corresponding to each theme.

Theme	Code	Interview of Participants (Participants’s Institution)
Secondary wounds	Check of wound surface	When the dressing is applied, the wound is not visible and the infection cannot be evaluated. (H3) I want to remove dressings temporarily. (H4)
	Easy to peel off/clean	I want to reduce the pain when (I) remove (the products). (H1) The high adhesive products hurt the skin of elderly people. (H2, H3) Silicon tapes had a gool balance between removal and adhesion. (H4) Patients may complain of pain when washing the wound many times (HC) When the gauze stuck to the wound surface is removed, bleeding from the wound may occur. (HC)
	Range to apply	Sheet-type products affect normal skin around the wound. (H4) The topical products will drip immediately when a wound of supine patient is treated.(HC) The dressings are displaced by the pressure exerted when sitting. (HC)
Adaptability of products	Necrotic tissue removal	(I want to) dissolve and remove the slough. (H1) I want to remove necrotic tissue without debridement.(H3) Products that can remove sloughs. (H4)
	Exudate control	There are many dressings used for wounds with a lot of exudates. (H1) (WMPs) need to absorb exudate. (H4) Exudate that promotes wound healing is absorbed by gauze. (HC) I want to use dressings that absorb the exudate well and are exchanged only once a week. (H2)
	Narrow pocket/Deep wound	It is difficult that topical products are applied to the pocket which remains during treatment. (H1) I want a thin spray nozzle so that medicine can be sprayed even if the entrance to the wound is narrow. (H2) I want products that can be fillable in a deep wound. (H4)
	Prevention	I want to prevent MDRPU under insurance coverage. (H4) For prevention of skin tears etc., I want to use dressings. But these are not under insurance coverage. (H4)
	Infection	Types of products used for infection wound are limited. (H2)
	Wide stage	"REF-THERA is moderately water-absorbing, can granulate, epithelialize and control infection, it is preferable to use.(HC) I select products with wide ranges of adaptation. (H1)
Convenience	Formulation	Spray type is difficult to use. (H1, H4) Easy to use if a medicine is solid. (H1,H4) I think that the product which is individually packaged in solid form is convenient. (H1) The adhesive surfaces stick together when they are applied. (HC) A drug-containing hydrogel sheet that is just applied is easy and can be used by the patient/family. (HC) Geben^®^ cream and pastes are hard and cannot be used in the state of being taken out of the tube. (H2)
	User	A product that can be used uniformly by all users is desired. (H1) A drug-containing hydrogel sheets that are just applied are easy and can be used by the patient/family. (HC) Elderly people who could not use the Internet or telephone have difficulty in ordering dressings. (H2) It is difficult to wash away the products from the viewpoint of the nurse. (H3) Macrogol base such as Isodine^®^ gel is easy to wash. (H1)
	Usage	The extension of the treatment time and the increase in the number of treatments impose a burden on the patient. (H1,H3,HC)
	Preservation and management	I want to keep the products at the patient’s bedside. (H1) The products stored in a cold place are often troublesome and wasteful. (H4) There are many patients who leave products in warm and humid place. (HC)
Economy of healthcare	Expiration date	Short-lived products such as Fiblast^®^ spray were often discarded. (H1)
	Apply again	I want repeatedly use dressings which are removed temporary. (H3) It is costly to apply the (new) dressings again. (H3) I want to move dressings temporarily. (H4) Presently, dressings are exchanged two or three times, which is costly. (H2)
	Inexpensive	Even if (a product is) inexpensive, it will be adopted if it is effective. (H1) The reason is that the dressings are inexpensive. (H4) Many families choose ointment and gauze rather than buying dressings twice a week for cost reasons. (H2)
	Standard of adoption/use	I select products with a wide range of application. (H1) I don’t use the different dressings (for each wound). (H4) The number of products that can be adopted in management is limited. (H2)
	Insurance coverage	I want to use for the purpose of prevention for MDRPU under insurance coverage. (H4) For prevention of skin tears etc., I want to use dressings. But these are not under insurance coverage. (H4)
Physicochemical properties	Thickness and hardness	Geben^®^ cream and pastes are hard and cannot be used in the state of being taken out of the tube. (H2) (Dressings are) as thin as possible. (H3, HC) U-PASTA© ointments are soft and usable. (H1) If the sacral dressings are too soft (decompressive) when sitting, it may be okay if there is a thickness. (HC) If the bases are too soft, (I thought that) misalignment may occur and the granulation may be damaged. (H2) If (the base is) too hard, (I thought that) it may hit the bones and cause friction. (HC)
	Size adjustment	I want dressings that can be cut such as film. (H4) I want dressings that contact the wound surface such as any large PU. (H2)
	Maintenance of form	The topical products will drip immediately when a wound of supine patient is treated. (HC) I want the product to fit a highly mobile sites such as neck. (HC) The dressings are displaced by the pressure exerted when sitting. (HC) The dressings do not misalignment even if a load such as body pressure is applied. (H4)
	Adhesion	The adhesiveness (of products) which are not misaligned does not burden the skin. (H4) Silicon tapes had a good balance between peeling and adhesion. (H4) If adhesion (of dressings) is weak, (the dressing is) easy to remove. (HC)
Patient QOL	Number of treatments	Productions with more than four daily exchanges have an impact on patient QOL. (HC) Patients may complain of pain, when washing the wound many times (HC)
	Leakage of topical product	The patient feels embarrassed to have their clothes soiled due to topical product leakage. (HC)

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
