# Peer review of "Preliminary Investigation of the Usability Characteristics Required for Wound Management Products by Semi-Structural Interview of Medical Staff"

_pharmacy, 2020, doi:10.3390/pharmacy8030152_

Round 1

Reviewer 1 Report

The purpose of this study was to conduct qualitative research on a few health professionals in a number of settings engaging in PU treatment to evaluate their needs regarding the usability of WMPs before conducting a massive (and it is assumed, quantitative) study of healthcare professions in this regard.

Although the authors say upfront that they have conducted this research on only a few persons at more than one site, there is no clear information on how many people were interviewed in total.  This matters because in the results section the authors often use the plural to say that participants thought one thing or another when it may be that it was only one person who actually thought something in particular.  If this is the case, a justification should be provided by the authors for why if only one healthcare professional brought up a point this information should be taken into consideration for this study.  It may very well be reasonable to include this information, but the reader needs to know why.

The authors have pointed out that they transcribed the information verbatim.  However, many of the points made, as indicated in the chart, are very difficult to understand.  I have guessed at what they might mean in my suggested edits below.  For the ones that are difficult to interpret, I wonder if the authors can know for sure what the healthcare professionals actually did intend?  Perhaps because they met the healthcare professionals they do know.  However, something needs to be said about how it is that the  researchers knew what the healthcare professionals meant when a number  of the statements presented in the chart are so unclear.  Below are my suggested edits line by line.

40 Change “can be applied family caregivers” to “can be applied by family caregivers”.

44 Change “when changes them” to “when a health care professional changes them” or to “when changing them”.

77 The total number of people participating in the study is unclear.  The total participants needs to be mentioned.

79  Even though the comments were transcribed verbatim, the wording of those comments needs to be understandable in the chart.  For this reason, I offer a number of changes to the wording to make it understandable.

82 Change “have you been treated” to “have you treated” or to “have you been treating”.

85-93 Change justification of text from centered to fully justified.

88 Change “indicating the content” to “the content”.

90 Change “levels” to “level”.

99 Change “Themes regarding the specific wound that a wound management products” to “Themes regarding the specific wound that a wound management product”.

Change “Themes regarding QOL of patient who continue to treatment” to “Themes regarding QOL of patients who continue treatment”.

100 This table needs to be redone.  The type is much too small.  Given that the text does not reach either the left of the right margin, the size of the text can easily be increased without changing the layout of the table.  The following are the changes suggested for the content of the table.

Change “wound becomes invisible” to “wound is not visible”

Change “I want to be removed dressing temporarily” to “I want to remove the dressing temporarily”.

Change “”high adhesive products are hurt” to “high adhesive products hurt”.

Change “many dressing used for wound with” to “many dressings used for wounds with”

Change “difficult that topical products applied to the pocket which remains” to “difficult that topical products are applied to the pocket which remains” or “difficult that topical products applied to the pocket remain”.

It is very hard to read the title of the section “Convenience”.  Check to make sure it is spelled correctly.

Change “easy to use” to “Easy to use”

Change “individually packaged with solid” to individually packaged in solid form”

Change “pastas” to “pastes”.

Change “have difficulty to order” to “have difficulty in ordering”.

Change “from viewpoint” to “from the viewpoint”.

Change “in the patient’s bedside” to “at the patient’s bedside”.

Change “use repeatedly dressings which removed temporary” to “repeatedly use dressings which are removed temporarily”.

Change “It costs to apply” to “It is costly to apply”.

Change “I want to be remove” to “I want to remove”.

Change “range of adaption” to “range of application”.

Change the title “Physiochemical propertis” to “Physiochemical properties”.

Change “pastas” to “pastes”.

Change “I want to the dressings” to “I want dressings”.

Change “I want to dressing that can be contacted the” to “I want dressings that contact”.

Change the heading “Maintain of form” to “Maintenance of form”

Change “fit in a highly” to “fit a highly”.

Change “The adhesiveness (of products) whici is not misalignment the form and is not burdens the skin” to “The adhesiveness (of products) which are not misaligned does not burden the skin”.

Change “Silicon tapes had a gool balance between peelin” to “Silicon tapes had a good balance between peeling”.

Change heading “Number of treatment” to “Number of treatments”.

It is very hard to read, but make sure the line says “Productions with more than four daily exchanges have an impact on patient QOL (HC)”

Change “patient feel embarrassed” to “patient feels embarrassed”.

106 Change “and “range of application”” to “and “range of application.””

215 I am not sure that the conclusion “This suggests that even professionals can have trouble selecting products.” follows from the participants commenting on the wide-range, similarity and difficulty in choosing.  The professionals may be making this comment from the perspective of the patient, thinking the patient may have difficulty in selecting.”

252 What does this mean? “In Japan, 1.6-50 % cases of MDRPUs occur in hospitals.” There is a very large difference between 1.6% and 50%.  Are you sure that this is the range you intended?

315 Change “tissue?.” to “tissue?”

Author Response

Dear Reviewer #1,

Thank you for giving us many suggestions.

We agree with your suggestion, and revised this manuscript thoroughly according to your suggestion.

  1. Although the authors say upfront that they have conducted this research on only a few persons at more than one site, there is no clear information on how many people were interviewed in total.  This matter because in the results section the authors often use the plural to say that participants thought one thing or another when it may be that it was only one person who actually thought something in particular.  If this is the case, a justification should be provided by the authors for why if only one healthcare professional brought up a point this information should be taken into consideration for this study.  It may very well be reasonable to include this information, but the reader needs to know why.
  • Thank you for suggestion. We added the interviewed healthcare professionals’ number in this manuscript.

  1. For the ones that are difficult to interpret, I wonder if the authors can know for sure what the healthcare professionals actually did intend?  Perhaps because they met the healthcare professionals they do know. 
  • We chose the clinic and hospitals from neighborhood. However, we interviewed healthcare professionals whom we actually haven’t known. Then, we used the interview guide, and did along it when we interviewed them. We are sure that we understand what they intend.

  1. Details

Line 40:  Change “can be applied family caregivers” to “can be applied by family caregivers”.

  • We revised as you suggested.

Line 44:  Change “when changes them” to “when a health care professional changes them” or to “when changing them”.

  • We revised as you suggested.

Line 77:  The total number of people participating in the study is unclear.  The total participants needs to be mentioned.

  • We revised as you suggested.

L 79: Even though the comments were transcribed verbatim, the wording of those comments needs to be understandable in the chart.  For this reason, I offer a number of changes to the wording to make it understandable.

L 82:  Change “have you been treated” to “have you treated” or to “have you been treating”.

L 85-93:  Change justification of text from centered to fully justified.

L 88:  Change “indicating the content” to “the content”.

L 90:  Change “levels” to “level”.

L 99:  Change “Themes regarding the specific wound that a wound management products” to “Themes regarding the specific wound that a wound management product”.

Change “Themes regarding QOL of patient who continue to treatment” to “Themes regarding QOL of patients who continue treatment”.

L 100:  This table needs to be redone.  The type is much too small.  Given that the text does not reach either the left of the right margin, the size of the text can easily be increased without changing the layout of the table.  The following are the changes suggested for the content of the table.

Change “wound becomes invisible” to “wound is not visible”

Change “I want to be removed dressing temporarily” to “I want to remove the dressing temporarily”.

Change “”high adhesive products are hurt” to “high adhesive products hurt”.

Change “many dressing used for wound with” to “many dressings used for wounds with”

Change “difficult that topical products applied to the pocket which remains” to “difficult that topical products are applied to the pocket which remains” or “difficult that topical products applied to the pocket remain”.

It is very hard to read the title of the section “Convenience”.  Check to make sure it is spelled correctly.

Change “easy to use” to “Easy to use”

Change “individually packaged with solid” to individually packaged in solid form”

Change “pastas” to “pastes”.

Change “have difficulty to order” to “have difficulty in ordering”.

Change “from viewpoint” to “from the viewpoint”.

Change “in the patient’s bedside” to “at the patient’s bedside”.

Change “use repeatedly dressings which removed temporary” to “repeatedly use dressings which are removed temporarily”.

Change “It costs to apply” to “It is costly to apply”.

Change “I want to be remove” to “I want to remove”.

Change “range of adaption” to “range of application”.

Change the title “Physiochemical propertis” to “Physiochemical properties”.

Change “pastas” to “pastes”.

Change “I want to the dressings” to “I want dressings”.

Change “I want to dressing that can be contacted the” to “I want dressings that contact”.

Change the heading “Maintain of form” to “Maintenance of form”

Change “fit in a highly” to “fit a highly”.

Change “The adhesiveness (of products) which is not misalignment the form and is not burdens the skin” to “The adhesiveness (of products) which are not misaligned does not burden the skin”.

Change “Silicon tapes had a gool balance between peelin” to “Silicon tapes had a good balance between peeling”.

Change heading “Number of treatment” to “Number of treatments”.

It is very hard to read, but make sure the line says “Productions with more than four daily exchanges have an impact on patient QOL (HC)”

Change “patient feel embarrassed” to “patient feels embarrassed”.

  • Change “and “range of application”” to “and “range of application.””
  •  
  • We are very sorry that there are many misspelling, and thank you very much for giving us many valuable suggestions.

We revised all as you suggested.

Especially, we re-tabulated the Table 3 which was very unclear, then the table was separated to 2 tables as Table 3(a) and (b).

  1. I am not sure that the conclusion “This suggests that even professionals can have trouble selecting products.” follows from the participants commenting on the wide-range, similarity and difficulty in choosing.  The professionals may be making this comment from the perspective of the patient, thinking the patient may have difficulty in selecting.”
  • We agree your comments. We would like to rewrite this sentence to

 “This suggests that even medical staffs who have profound knowledge about the care for PU find it difficult to select the appropriate products.” At Line 240-241 in revised manuscript,

252 What does this mean? “In Japan, 1.6-50 % cases of MDRPUs occur in hospitals.” There is a very large difference between 1.6% and 50%.  Are you sure that this is the range you intended?

  • I'm sorry that this sentence was confused. We revised this sentence at line 278-279 as “In Japan, PUs caused by MDRPUs have been reported at various occurrence rates such as 1.6% at general hospital and 50% at pediatric hospital.” At line 278-289 in revised manuscript.

315 Change “tissue?.” to “tissue?”

  • We revised as you suggested.

Reviewer 2 Report

Thank you for the opportunity to review your manuscript.  I agree that asking health professionals working with wounds what the best product characteristics are could lead to improved products.

There are a few comments I would like to make, I am hopeful that they will help you to produce a manuscript that you will be proud to have published.

The manuscript title is problematic on several levels 

*I don't know what a "Massive Study" is.

*There is no mention of a follow-up study

*This seems to be almost self-promotion - the title seems to say, "Here is our initial work, but something bigger is coming next."

*"to medical staff" doesn't make sense

Please consider amending the title to remove this.  Perhaps "Preliminary investigation of the usability characteristics required for wound management products by semi-structural interview of medical staff"

Line 31:  "originate" is not the best word.  Chronic wounds do become infected, but they do not "originate" the infectious organism.

Line 40:  I believe there are 2 missing words - consider "can be applied BY family AND caregivers..."

Line 41:  Aren't dressings "topical products"?  Is there another term or phrase to describe the difference between dressings and other topical products?

Lines 43-44:  I am not familiar with Iodocoat, but this description is hard to understand.  Iodocoat absorbes exudates into a gel matrix, then what?  If there is no next-step, this sentence should end with matrix.  Is it the formation of the gel matrix that is useful?  When who changes them?  Are all changes done "at bed side"?

Line 50: there should not be a space between 90 and %

Line 68: again the phrase "massive study". You don't "make" a study, you may "conduct" a study or "administer" a survey.  Generally the phrase is study "of" medical staff.  I don't know anything about the follow-up study, but I am hopeful that caregivers were included.  If they were, they should be included in this sentence.

Table 1: this is a non-traditional spelling of pharmacist

Table 1: when listing ranges, it is common to use a hyphen (-). I do not understand the choice of (~). is that to indicate an approximation?  If so, it usually does not appear in the middle of the range.

Table 1: is a stoma truly a chronic wound?  I understand it is an opening in the skin, but a stoma is intended to be permanent and is rarely considered to be a wound.

Line 91:  I do not understand the spacing on this line

Figure 1:  This is very well presented, thank you

Line 273:  A brief expansion on reaching theoretical saturation would be helpful, either here or in the methods section.

Author Response

Dear Reviewer #2,

Thank you for reviewing and making suggestions.

According to your suggestions, we revised this manuscript as below.

  1. *I don't know what a "Massive Study" is.
  2. *There is no mention of a follow-up study
  3. *This seems to be almost self-promotion - the title seems to say, "Here is our initial work, but something bigger is coming next."
  4. *"to medical staff" doesn't make sense
  5. Please consider amending the title to remove this.  Perhaps "Preliminary investigation of the usability characteristics required for wound management products by semi-structural interview of medical staff"
  • Thank you for your suggestion. That’s right, then we changed the title as you suggested.

Line 31:  "originate" is not the best word.  Chronic wounds do become infected, but they do not "originate" the infectious organism.

  • Thank you for your suggestion. We revised the words to “trigger”.

Line 40:  I believe there are 2 missing words - consider "can be applied BY family AND caregivers..."

  • Thank you for your suggestion. We revised as you suggested.

Line 41:  Aren't dressings "topical products"?  Is there another term or phrase to describe the difference between dressings and other topical products?

  • Thank you for your suggestion. We revised as you suggested.

Lines 43-44:  I am not familiar with Iodocoat, but this description is hard to understand.  Iodocoat absorbes exudates into a gel matrix, then what?  If there is no next-step, this sentence should end with matrix.  Is it the formation of the gel matrix that is useful?  When who changes them?  Are all changes done "at bed side"?

  • Thank you for your suggestion. We revised as you suggested.

Line 50: there should not be a space between 90 and %

  • Thank you for your suggestion. We revised as you suggested.

Line 68: again the phrase "massive study". You don't "make" a study, you may "conduct" a study or "administer" a survey.  Generally the phrase is study "of" medical staff.  I don't know anything about the follow-up study, but I am hopeful that caregivers were included.  If they were, they should be included in this sentence.

  • Thank you for your suggestion. We actually did interview health professionals as preliminary investigation before massive study to all people involving wound care in this study. So, we deleted about the description in the manuscript.

Table 1: this is a non-traditional spelling of pharmacist

Table 1: when listing ranges, it is common to use a hyphen (-). I do not understand the choice of (~). is that to indicate an approximation?  If so, it usually does not appear in the middle of the range.

Table 1: is a stoma truly a chronic wound?  I understand it is an opening in the skin, but a stoma is intended to be permanent and is rarely considered to be a wound.

Line 91:  I do not understand the spacing on this line

  • Thank you for your suggestion. We revised all in the manuscript.

Figure 1:  This is very well presented, thank you

  • We are very happy to your comment. Thank you very much.

Line 273:  A brief expansion on reaching theoretical saturation would be helpful, either here or in the methods section.

  • Thank you for your suggestion. We added about the theoretical saturation in the methods.
